# Learning Antidote Data to Individual Unfairness

## Abstract

Fairness is an essential factor for machine learning systems deployed in high-stake applications. Among all fairness notions, individual fairness, following a consensus that 'similar individuals should be treated similarly,' is a vital notion to guarantee fair treatment for individual cases. Previous methods typically characterize individual fairness as a prediction-invariant problem when perturbing sensitive attributes, and solve it by adopting the Distributionally Robust Optimization (DRO) paradigm. However, adversarial perturbations along a direction covering sensitive information do not consider the inherent feature correlations or innate data constraints, and thus mislead the model to optimize at off-manifold and unrealistic samples. In light of this, we propose a method to learn and generate antidote data that approximately follows the data distribution to remedy individual unfairness. These on-manifold antidote data can be used through a generic optimization procedure with original training data, resulting in a pure pre-processing approach to individual unfairness, or can also fit well with the in-processing DRO paradigm. Through extensive experiments, we demonstrate our antidote data resists individual unfairness at a minimal or zero cost to the model's predictive utility.

## 1 Introduction

Unregulated decisions could reflect racism, ageism, and sexism in high-stakes applications, such as grant assignments (Mervis, 2022), recruitment (Dastin, 2018), policing strategies (Gelman et al., 2007), and lending services (Bartlett et al., 2022). To avoid societal concerns, fairness, as one of the fundamental ethical guidelines for AI, has been proposed to encourage practitioners to adopt AI responsibly and fairly. The unifying idea of fairness articulates that ML systems should not discriminate against individuals or any groups segmented by legally-protected and sensitive attributes, therefore preventing disparate impact in automated decision-making (Barocas & Selbst, 2016).

Many notions have been proposed to specify AI Fairness (Dwork et al., 2012; Kusner et al., 2017; Hashimoto et al., 2018). Group fairness is currently the most influential notion in the fairness community, driving different groups to receive equitable outcomes regardless of their sensitive attributes, in terms of statistics like true positive rates or positive rates (Hardt et al., 2016). However, these statistics describe the average of a group, hence lacking guarantees on the treatments of individual cases. Alternatively, individual fairness established upon a consensus that 'similar individuals should be treated similarly,' shift force to reduce the predictive gap between conceptually similar instances. Here, 'similar' means two instances have close profiles regardless of their different sensitive attributes, and usually have customized definitions upon domain knowledge. We invite readers to look into Section 2 for a more concrete establishment on individual fairness.

Previous methods solve the individual fairness problem mainly by Distributionally Robust Optimization (DRO) (Yurochkin et al., 2020; Yurochkin & Sun, 2021; Ruoss et al., 2020; Yeom & Fredrikson, 2021). They convert the problem to optimize models for invariant predictions towards original data and their perturbations, where the perturbations are adversarially constructed to mostly change the sensitive information in a sample. However, one use case of DRO in model robustness is to adversarially perturb a sample by a small degree. The perturbations can be regarded as local perturbations, and the adversarial sample is still on the data manifold. In contrast, perturbing a sample for individual fairness purposes, *e.g.*, directly flipping its sensitive attributes like gender from male to female, cannot be regarded as a local perturbation. These perturbations may violate inherent feature cor-

relations, *e.g.*, some features are subject to gender but without notice, thus driving the adversarial samples leaving the data manifold. Additionally, perturbations in a continuous space could break the innate constraints from tabular, *e.g.*, discrete features should be in a one-hot format. Consequently, these adversarial samples for fairness are unrealistic and do not match the data distribution. Taking these data can result in sub-optimal tradeoffs between utility and individual fairness.

In this work, we address the above limitations and propose an approach to rectify models for individual fairness from a pure data-centric perspective. Following the high-level idea of the DRO paradigm, and by giving a concrete setup for similar samples, we learn the data manifold through generative models, and continue to construct on-manifold samples with different sensitive attributes as *antidote data* to mitigate individual unfairness. We launch two ways to use the generated antidote data: simply inserting antidote data into the original training set and training models through regular optimization, or equipping antidote data to the DRO pipeline as an in-processing approach. Our approach works for multiple sensitive attributes, and each sensitive attribute can have multiple values. We conduct experiments on census, criminological, and educational datasets, compared to standard classifiers and several baseline methods. Compared to baseline methods, our method greatly mitigates individual unfairness, and has minimal or zero side effects to model utility.

## 2 INVIDIVUAL FAIRNESS AND COMPARABLE SAMPLES

**Notations**  Let $f_\theta$ denote a parameterized probabilistic classifier, $\mathcal{X}$ and $\mathcal{Y}$ denote input and output space with instance $x$ and label $y$, respectively. For tabular datasets, we assume every input instance $x$ contains three parts of features: sensitive features $\mathbf{s} = [s_1, s_2, \cdots, s_{N_s}]$, continuous features $\mathbf{c} = [c_1, c_2, \cdots, c_{N_c}]$, and discrete features $\mathbf{d} = [d_1, d_2, \cdots, d_{N_d}]$, with $N$ denoting the number of features in each parts. We assume these three parts of features are exclusive, *i.e.*, $\mathbf{s}$, $\mathbf{c}$, and $\mathbf{d}$ do not share any feature or column. We use $\mathbf{d}_x$ to denote the discrete features of instance $x$, and the same manner for other features. For simplification we shall assume discrete features $\mathbf{d}$ contain categorical features before one-hot encoding, continuous features $\mathbf{c}$ contain features in a unified range like $[0, 1]$ after some scaling operations, and all data has the same feature dimension. We consider sensitive attributes in a categorical format. Any continuous sensitive attribute can be binned into discrete intervals to fit our scope. We use $\oplus$ to denote vector-vector or vector-scalar concatenation.

**Individual Fairness: Concept and Practical Usage**  The concept of individual fairness is firstly raised in Dwork et al. (2012). Following a consensus that 'similar individuals should be treated similarly,' the problem is formulated as a Lipschitz mapping problem. Formally, for arbitrary instances $x$ and $x' \in \mathcal{X}$, individual fairness is defined as a $(D_\mathcal{X}, D_\mathcal{Y})$-Lipschitz property of a classifier $f_\theta$:

$$D_\mathcal{Y}(f_\theta(x), f_\theta(x')) \leq D_\mathcal{X}(x, x'), \tag{1}$$

where $D_\mathcal{X}(\cdot, \cdot)$ and $D_\mathcal{Y}(\cdot, \cdot)$ are some distance functions respectively defined in the input space $\mathcal{X}$ and output space $\mathcal{Y}$, and shall be customized upon domain knowledge. However, for a general problem, it could be demanding to carry out a concrete and interpretable $D_\mathcal{X}(\cdot, \cdot)$ and $D_\mathcal{Y}(\cdot, \cdot)$, hence makes individual fairness impractical in many applications. To simplify this problem from a continuous Lipschitz constraint, some works evaluate individual fairness of models with a binary distance function: $D_\mathcal{X}(x, x') = 0$ for two different samples $x$ and $x'$ if they are exactly the same except sensitive attributes, *i.e.*, $\mathbf{c} = \mathbf{c}'$, $\mathbf{d} = \mathbf{d}'$, and $\mathbf{s} \neq \mathbf{s}'$ (Yurochkin et al., 2020; Yurochkin & Sun, 2021). Despite the interpretability, this constraint can be too harsh to find sufficient comparable samples since other attributes may correlate with sensitive attributes. For empirical studies, these studies can only simulate the experiments with semi-synthetic data where they flip one's sensitive attribute to construct a sample and evaluate the predictive gap. Note that for tabular data, simply discarding the sensitive attributes could be a perfectly individually fair solution to this simulation.

In this work, we consider a relaxed version of the above individual fairness definition for an imperfect classifier. We present Definition 2.1 to characterize in what conditions we shall consider two samples are *comparable*. When two samples $x$ and $x'$ are coming to be *comparable*, their predictive gap $|f_\theta(x) - f_\theta(x')|$ should be minimized for the individual fairness purpose.

**Definition 2.1** (*comparable samples*). Given $T_d, T_c \in \mathbb{R}_{\geq 0}$, $x$ and $x'$ are *comparable* iff all constraints are satisfied: 1. $\sum_{i=1}^{N_d} \mathbb{1}\{d_i \neq d_i'\} \leq T_d$; 2. $\max\{|c_i - c_i'|\} \leq T_c, \forall\, 1 \leq i \leq N_c$; and 3. $y = y'$.

*Remark* 2.1.  For some thresholds $T_d$ and $T_c$, two samples are considered as *comparable* iff 1. there are at most $T_d$ features differing in discrete features; 2. the largest disparity among all continuous features is smaller or equal to $T_c$, and 3. two samples have the same ground-truth label.

Definition 2.1 allows two samples to be slightly different in discrete and continuous features, and arbitrarily different in sensitive attributes. The definition is also flexible to extend if users want to enforce some crucial features to be identical for *comparable samples*, and this does not affect our model design. Our *comparable samples* are highly interpretable and semantically rich. For example, in lending data, to certify individual fairness for two samples, we can set discrete features to the history of past payment status (where value 1 indicates a complete payment, and value 0 indicates a missing payment), and continuous features to the monthly amount of bill statement. Two samples are considered to be *comparable* if they have a determinate difference in payment status and amount of bills. In what follows we shall build models and evaluate individual fairness by Definition 2.1, and mostly consider *comparable samples* with different sensitive attributes.

## 3    Learning Antidote Data to Individual Unfairness

**Motivation**    Several methods solve the individual fairness problem through Distributionally Robust Optimization (DRO) (Yurochkin et al., 2020; Yurochkin & Sun, 2021; Ruoss et al., 2020; Yeom & Fredrikson, 2021). The high-level idea is to optimize a model at some samples with perturbations that dramatically change their sensitive information. The solution can be summarized as:

$$\min_{f_\theta} \mathbb{E}_{(x,y)} \ell(f_\theta(x), y) \quad \text{and} \quad \min_{f_\theta} \mathbb{E}_{(x,y)} \max_{x+\epsilon \sim \mathcal{D}_{\text{Sen}}} \ell(f(x+\epsilon), y), \tag{2}$$

where the first term is standard empirical risk minimization, and the second term is for loss minimization over adversarial samples. $\mathcal{D}_{\text{Sen}}$ is some customized distribution offering perturbations to specifically change one's sensitive information. For example, Yurochkin et al. (2020) characterizes $\mathcal{D}_{\text{Sen}}$ as a subspace called sensitive subspace learnt from logistic regression, which contains the most predictability of sensitive attributes. Ruoss et al. (2020) find out this distribution via logical constraints. Though feasible, we would like to respectfully point out that (1) Perturbations violate feature correlations could push adversarial samples leave the data manifold. An intuitive example is treating age as a sensitive attribute. Perturbations can change a person's age arbitrarily to find an optimal age that encourage the model to predict the most differently. Such perturbations ignore the correlations between the sensitive feature and other features like education or annual income, resulting in an adversarial sample with age 5 or 10 but holding a doctoral degree or getting $80K annual income. (2) Samples with arbitrary continuous perturbations can easily break the nature of tabular data. There are only one-hot discrete values for categorical variables after one-hot encoding, and potentially a fixed range for continuous variables. For example, the adversarial samples may in half bachelor degree and half doctoral degree. These two observations make the adversarial samples from $\mathcal{D}_{\text{Sen}}$ unrealistic and leaving the data manifold, thus distorting the following DRO paradigm, and resulting in sub-optimal tradeoffs between fairness and utility.

In this work, we address the above issues related to $\mathcal{D}_{\text{Sen}}$, and propose to generate on-manifold data for individual fairness purposes. The high-level philosophy is, by giving an original training sample, generate its *comparable samples* with different and reasonable sensitive attributes, and the generated data should fit into existing data manifold and obey the inherent feature correlations or innate data constraints. We name the generated data as *antidote data*. The *antidote data* can either mix with original training data to be a pre-processing technique, or either serve as $\mathcal{D}_{\text{Sen}}$ in Equation (2) as an in-processing approach. By taking *antidote data*, a classifier would give individually fair predictions.

### 3.1    Antidote Data Generator

We start by elaborating on the generator of antidote data. The purpose of antidote data generator $g_\theta$ is, given a training sample $x$, generating its comparable samples with different sensitive attribute(s). To ensure the generations have different sensitive features, we build $g_\theta$ as a conditional generative model to generate a sample with pre-defined sensitive features. Given sensitive attributes $\bar{\mathbf{s}} \neq \mathbf{s}_x$ (recall $\mathbf{s}_x$ is the sensitive attributes of instance $x$), the objective is:

$$g_\theta : (x, \bar{\mathbf{s}}, \mathbf{z}) \to \hat{x}, \quad \text{with} \quad \mathbf{s}_{\hat{x}} = \bar{\mathbf{s}}, \quad x \text{ and } \hat{x} \text{ satisfy Definition 2.1}, \tag{3}$$

where $\mathbf{z} \sim \mathcal{N}(0, 1)$ is drawn from a standard multivariate normal distribution as a noise vector. The generation $\hat{x}$ should follow the data distribution and satisfy some innate constraints from discrete or continuous features, *i.e.*, the one-hot format for discrete features and a reasonable range for continuous features. In the following, we shall elaborate the design and training strategy for $g_\theta$.

**Encoding Continuous Values** For continuous features, we adopt mode-specific normalization (Xu et al., 2019) to encode every column of continuous values independently. We use Variational Bayesian to estimate the Gaussian mixture in the distribution of one continuous feature. This approach will decompose the distribution into several modes, where each mode is a Gaussian distribution with unique parameters. Formally, given a value $c_{i,j}$ in the $i$-th column of continuous feature and $j$-th row in the tabular, the learned Gaussian mixture is $\mathbb{P}(c_{i,j}) = \sum_{k=1}^{K_i} w_{i,k} \mathcal{N}(c_{i,j}; \mu_{i,k}, \sigma_{i,k}^2)$, where $w_{i,k}$ is the weight of $k$-th mode in $i$-th continuous feature, and $\mu_k$ and $\sigma_k$ are the mean and standard deviation of the normal distribution of $k$-th mode. We use the learned Gaussian mixture to encode every continuous value. For each value $c_{i,j}$, we estimate the probability from each mode via $p_{i,k}(c_{i,j}) = w_{i,k} \mathcal{N}(c_{i,j}; \mu_{i,k}, \sigma_{i,k}^2)$, and sample one mode from the discrete probability distribution $\mathbf{p}_i$ with $K_i$ values. Having a sampled mode $k$, we represent the mode of $c_{i,j}$ using a one-hot mode indicator vector, an all-zero vector $\mathbf{e}_{i,x}$ except the $k$-th entry equal to 1. We use a scalar to represent the relative value within $k$-th mode: $v_{i,x} = (c_{i,j} - \mu_{i,k})/4\sigma_{i,k}$. By encoding all continuous values, we have a re-representation $\tilde{x}$ to substitute $x$ as as the input for antidote data generator $g_\theta$:

$$\tilde{x} = (v_{1,x} \oplus \mathbf{e}_{1,x} \oplus \cdots \oplus v_{N_c,x} \oplus \mathbf{e}_{N_c,x}) \oplus \mathbf{d}_x \oplus \mathbf{s}_x. \tag{4}$$

Recall $\oplus$ denotes vector-vector or vector-scalar concatenation. To construct a comparable sample $\hat{x}$, the task for continuous features is to classify the mode from latent representations, *i.e.*, estimate $\mathbf{e}_{i,k}$, and predict the relative value $v_{i,x}$. We can decode $v_{i,x}$ and $\mathbf{e}_{i,x}$ back to a continuous value using the learned Gaussian mixture.

**Structural Design** The whole model is designed in a Generative Adversarial Networks (Goodfellow et al., 2014) style, consisting of a generator $g_\theta$ and a discriminator $d_\theta$.

The generator $g_\theta$ takes the re-representation $\tilde{x}$, a pre-defined sensitive feature $\bar{\mathbf{s}}$, and noisy vector $\mathbf{z}$ as input. The output from $g_\theta$ will be a vector with the same size as $\tilde{x}$ including $v_{\hat{x}}$, $\mathbf{e}_{\hat{x}}$, $\mathbf{d}_{\hat{x}}$, and $\mathbf{s}_{\hat{x}}$. To ensure all discrete features are in a one-hot manner so that the generations will follow a tabular distribution, we apply Gumbel softmax (Jang et al., 2017) as the final activation to each discrete feature and obtain $\mathbf{d}_{\hat{x}}$. Gumbel softmax is a differentiable operation to encode a continuous distribution over a simplex and approximate it to a categorical distribution. This function controls the sharpness of output via a hyperparameter called temperature. Gumbel softmax is also applied to sensitive features $\mathbf{s}_{\hat{x}}$ and mode indicator vectors $\mathbf{e}_{\hat{x}}$ to ensure the one-hot format.

The purpose for the discriminator model $d_\theta$ is to distinguish the fake generations from real samples, and we also build discriminator to identify generated samples in terms of its comparability from real comparable samples. Through discriminator, the constraints from *comparable samples* are implicitly encoded into the adversarial training. We formulate the fake sample for discriminator as $\hat{x} \oplus \tilde{x} \oplus (\hat{x} - \tilde{x})$, and real samples as $\tilde{x}' \oplus \tilde{x} \oplus (\tilde{x}' - \tilde{x})$, where $\tilde{x}'$ is the re-representation of a comparable sample $x'$ to $x$ drawn from the training data. The third term $\hat{x} - \tilde{x}$ is encoded to emphasize the different between two comparable samples. Implicitly regularizing the comparability leaves full flexibility to the generator to fit with various definitions of comparable samples, and avoid adding complicated penalty terms, as long as there are real comparable samples prepared for training.

**Training Antidote Data Generator** We train generator and discriminator iteratively through the following objectives with gradient penalty Gulrajani et al. (2017) to ensure stability:

$$\min_{g_\theta} \mathbb{E}_{x,x' \sim \mathcal{D}_{\text{comp}}} \quad \ell_{\text{CE}}(\mathbf{s}_{\hat{x}}, \mathbf{s}_{x'}) - d_\theta(g_\theta(\tilde{x} \oplus \mathbf{s}_{x'} \oplus \mathbf{z})), \tag{5}$$

$$\min_{d_\theta} \mathbb{E}_{x,x' \sim \mathcal{D}_{\text{comp}}} \quad d_\theta(g_\theta(\tilde{x} \oplus \mathbf{s}_{x'} \oplus \mathbf{z})) - d_\theta(\tilde{x}'), \tag{6}$$

where $\mathcal{D}_{\text{comp}}$ is the distribution describing the real comparable samples in data, $\ell_{\text{CE}}$ is cross entropy loss to penalty the prediction of every sensitive attribute in $\mathbf{s}_{\hat{x}}$ with $\mathbf{s}_{x'}$ as the ground-truth.

## 3.2 LEARNING WITH ANTIDOTE DATA

We elaborate two ways to easily apply the generated antidote data for the individual fairness purpose.

In practice, it is not strictly guaranteed that $g_\theta$ will produce comparable samples submitting to Definition 2.1. Some samples may be incompatible with some pre-defined sensitive features coming from the violations of neural networks. Thus, we apply a post-processing step `Post` to filter out comparable samples from all the generations. Given a dataset $X$, for one iteration of sampling, we input every $x$ with all possible sensitive features (except $\mathbf{s}_x$) to the generator, collect raw generations

---

**Algorithm 1** DRO-Anti: DRO with Antidote Data for Individual Fairness

---

1: **Input:** Training data $T = \{(x_i, y_i)\}^N$, learning rate $\eta$, loss function $\ell$
2: Train Comparable Sample Generator $g_\theta$ with $\{x_i\}^N$ and comparable constraints
3: Sample antidote data $\hat{X}$ using $g_\theta$
4: **repeat**
5: $\quad \theta \leftarrow \theta - \eta \mathbb{E}_{(x,y)}[\nabla_\theta(\max_{\hat{x} \in \{\hat{x}_i\}^M \leftarrow x} \ell(\hat{x}, y) + \ell(x, y))] \qquad \triangleright \{\hat{x}_i\}^M \leftarrow x$ is the set of $M$
    comparable samples of $x$ and $\{\hat{x}_i\}^M \in \texttt{Post}(\hat{X})$
6: **until** convergence

---

$\hat{X}$, and apply $\texttt{Post}(\hat{X})$ to get the antidote data. The label $y$ for antidote data is copied from the original data. We may have multiple iterations of sampling to enlarge the pool of antidote data.

The first way to use antidote data is to simply insert all antidote data to the original training set:

$$\min_{f_\theta} \sum \ell(f_\theta(x), y), \quad x \in X + \texttt{Post}(\hat{X}). \tag{7}$$

Since we only add additional training data, this approach is model-agnostic, flexible to any model optimization procedure, and fits well with well-developed data analytical toolkits such as sklearn (Pedregosa et al., 2011). We consider the convenience as a favorable property for practitioners.

The second way is to apply antidote data with Distributionally Robust Optimization. We present the training procedure in Algorithm 1. In every training iteration, except the optimization at real data with $\ell(x, y)$, we add an additional step to select $x$'s comparable samples in antidote data with the highest loss incurred by the current model's parameters, and capture gradients from $\max_{\hat{x} \in \{\hat{x}_i\}^M \leftarrow x} \ell(\hat{x}, y)$ to update the model. The algorithm is similar to DRO with perturbations along some sensitive directions, but instead we replace the perturbations with on-manifold generated data. The additional loss term in Algorithm 1 can be upper bounded by a gradient smoothing regularization term. Taking Taylor expansion, we have:

$$
\begin{aligned}
\max_{\hat{x} \in \{\hat{x}_i\}^m \leftarrow x} \ell(\hat{x}, y) &= \ell(x, y) + \max_{\hat{x} \in \{\hat{x}_i\}^m \leftarrow x} [\ell(\hat{x}, y) - \ell(x, y)] \\
&= \ell(x, y) + \max_{\hat{x} \in \{\hat{x}_i\}^m \leftarrow x} [\langle \nabla_x \ell(x, y), (\hat{x} - x) \rangle] + \mathcal{O}(\delta^2) \\
&\leq \ell(x, y) + T_d \max_i \nabla_{d_i} \ell(x, y) + T_c \max_i \nabla_{c_i} \ell(x, y) + N_s \max_i \nabla_{s_i} \ell(x, y) + \mathcal{O}(\delta^2).
\end{aligned}
\tag{8}
$$

Recall $T_d$ and $T_c$ are the thresholds for discrete and continuous features in Definition 2.1. ${\color{red}\mathcal{O}(\delta^2)}$ is the higher-order from Taylor expansion. The last inequality is from Definition 2.1. The three gradients on discrete, continuous, and sensitive features serve as gradient regularization and encourage the model to have invariant loss with regard to comparable samples. However, the upper bound is only a sufficient but not necessary condition, and our solution encodes real data distribution into the gradient regularization to solve individual unfairness with more favorable trade-offs.

## 4 EXPERIMENTS

### 4.1 EXPERIMENTAL SETUP

**Datasets** We involve censual datasets *Adult* (Kohavi & Becker, 1996) and *Dutch* (Van der Laan, 2000), educational dataset *Law School* (Wightman, 1998) and *Oulad* (Kuzilek et al., 2017), and criminological dataset *Compas* (Angwin et al., 2016) in our experiments. For each dataset, we select one or two attributes related to ethics as sensitive attributes which expose a significant individual unfairness in a base model like neural networks. We report their details in Appendix A.

**Protocol** For all datasets, we transform discrete features into one-hot encoding, and standardize the features by removing the mean and scaling to unit variance. We transform continuous features into the range between 0 and 1. We construct pairs of comparable samples for both training and testing sets. In experiments, different from (Yurochkin & Sun, 2021; Yurochkin et al., 2020), our evaluations on tabular datasets are sampled from real testing data but not simulated. We evaluate both the model utility and individual fairness in experiments. For utility, we consider the area under the Receiver Operating Characteristic Curve (ROC), and Average Precision (AP) to characterize the precision of probabilistic outputs in binary classification. For individual fairness, we consider

Table 1: Experimental results on *Adult* dataset

| | ROC ↑ | AP ↑ | Pos. Comp. (Mean/Q3) ↓ | Neg. Comp. (Mean/Q3) ↓ |
|---|---|---|---|---|
| LR (Base) | **90.04** | **75.72** | 31.75 / 43.55 | 10.25 / 18.37 |
| LR+Proj | 81.40 -9.60% | 62.19 -17.87% | 25.10 -20.95% / 34.83 -20.02% | 23.29 +127.17% / 33.03 +79.81% |
| LR+Dis | 89.95 -0.10% | 75.59 -0.17% | 30.81 -2.94% / 41.10 -5.62% | 9.40 -8.29% / 17.78 -3.18% |
| LR+Anti | 89.72 -0.35% | 75.04 -0.90% | 24.72 -22.13% / 30.84 -29.18% | 8.66 -15.56% / 14.64 -20.29% |
| LR+Anti+Dis | 89.56 -0.53% | 74.83 -1.17% | **23.02** -27.49% / **26.61** -38.90% | **8.12** -20.76% / **13.91** -24.28% |
| NN (Base) | **88.18** | 70.09 | 33.21 / 47.84 | 13.03 / 23.37 |
| NN+Proj | 87.42 -0.86% | 68.51 -2.25% | 32.38 -2.52% / 46.45 -2.91% | 13.59 +4.36% / 23.69 +1.37% |
| NN+Dis | 88.15 -0.04% | 70.27 +0.26% | 32.90 -0.93% / 44.79 -6.37% | 11.83 -9.17% / 23.36 -0.08% |
| SenSR | 86.01 -2.47% | 66.19 -5.57% | 28.68 -13.63% / 44.21 -7.59% | 14.88 +14.20% / 23.07 -1.31% |
| SenSeI | 86.42 -2.00% | 66.08 -5.72% | 27.92 -15.94% / 35.92 -24.91% | 13.22 +1.53% / 26.01 +11.28% |
| LCIFR | 87.35 -0.94% | 68.52 -2.24% | 32.51 -2.13% / 44.84 -6.26% | 12.97 -0.41% / 26.49 +13.35% |
| NN+Anti | 87.95 -0.26% | 69.51 -0.83% | 26.05 -21.57% / 35.76 -25.26% | 10.42 -19.97% / 16.88 -27.80% |
| NN+Anti+Dis | 87.79 -0.44% | 69.40 -0.98% | 24.40 -26.53% / 32.12 -32.85% | 9.56 -26.63% / 15.54 -33.51% |
| DRO-Anti | 87.91 -0.31% | **71.08** +1.41% | **17.46** -47.44% / **20.04** -58.10% | **5.48** -57.96% / **6.87** -70.59% |

the gap in probabilistic scores between comparable samples when both two samples have the same positive or negative label (abbreviated as Pos. Comp. and Neg. Comp.). We evaluate unfairness for positive and negative comparable samples in terms of the arithmetic mean (Mean) and upper quartile (Q3). The upper quartile can show us the performance of some worse-performed pairs. For a base model with randomness like NN, we ran the experiments five times and report the average results.

**Baselines** We consider two base models: logistic regression (**LR**), and three-layers neural networks (**NN**). We use logistic regression from Scikit-learn (Pedregosa et al., 2011), and our antidote data is compatible with this mature implementation since it does not make a change to the model. Approaches involving DRO currently do not support this LR pipeline, but will be validated through neural networks implemented with PyTorch. We have the following five baselines in experiments: 1. Discard sensitive features (**Dis**). This approach simply deletes the appointed sensitive features in the input data; 2. Project (**Proj**) (Yurochkin et al., 2020). Project finds a linear projection via logistic regression which minimizes the predictability of sensitive attributes in data. It requires an extra pre-processing step to project input data. 3. **SenSR** (Yurochkin et al., 2020). SenSR is based on DRO. It finds a sensitive subspace through logistic regression which encodes the sensitive information most, and generates perturbations on this sensitive subspace during optimization. 4. **SenSeI** (Yurochkin & Sun, 2021). SenSeI also uses the DRO paradigm, but involves distances penalties on both input and model predictions to construct perturbations; 5. **LCIFR** (Ruoss et al., 2020). LCIFR computes adversarial perturbations with logical constraints, and optimizes representations under the attacks from perturbations. We basically follow the default hyperparameter setting from the original implementation but fine-tune some parameters to avoid degeneration in some cases. For our approaches, we use **Anti** to denote the approach that simply merges original data and antidote data, use 'Anti.' to denote adding Dis to original and antidote data, and use **DRO-Anti** to denote antidote data with DRO. We standardized baselines with the same base model in experiments.

## 4.2 HOW ANTIDOTE DATA MITIGATE UNFAIRNESS

We present our numerical results on Table 1, Table 2, and Figure 1, and defer more to Appendix C. From these results we have the following major observations.

**Antidote Data Show Good Performance** Across all datasets, with antidote data, our models mostly perform the best in terms of individual fairness, and with only a minimal drop or sometimes even a slight improvement on predictive utility. For example, on *Law School* dataset, our NN+Anti mitigates individual unfairness by 70.38% and 63.36% in terms of the Mean in Pos. Comp. and Neg. Comp., respectively, with improvements on ROC by 0.47% and AP by 0.07%. On this dataset, other methods typically bring a 0.1%-2.5% drop in utility, and deliver less mitigation on individual unfairness. In some cases, some baseline methods do give better individual fairness, *e.g.*, LCIFR for Neg. Comp., but their fairness is not consistent for positive comparable samples, which is usually achieved at a significant cost on utility (up to a 13.03% drop in ROC).

**Improvements from DRO-Anti** Our DRO-Anti outperforms base models that learn with antidote data through regular optimizations. This model gets fairer results and slightly better predictive utility. This is because DRO-Anti introduces antidote data into every optimization iteration and selects the worst performed data instead of treating them equally. The typical DRO training has an

Table 2: Experimental results on *Law School* dataset

| | ROC ↑ | AP ↑ | Pos. Comp. (Mean/Q3) ↓ | Neg. Comp. (Mean/Q3) ↓ |
|---|---|---|---|---|
| LR (Base) | 86.14 | 97.80 | 3.67 / 5.39 | 11.70 / 15.21 |
| LR+Proj | 85.84 -0.35% | 97.74 -0.06% | 2.23 -39.35% / 2.48 -54.07% | 8.66 -25.99% / 11.40 -25.05% |
| LR+Dis | 86.18 +0.04% | 97.79 -0.01% | 2.04 -44.32% / 2.32 -56.90% | 7.33 -37.36% / 11.25 -26.04% |
| LR+Anti | **86.22** +0.08% | 97.80 +0.00% | 1.79 -51.20% / 2.20 -59.14% | 6.56 -43.98% / 8.64 -43.21% |
| LR+Anti+Dis | 86.20 +0.06% | **97.80** -0.00% | **1.76** -52.08% / **2.16** -59.96% | **6.28** -46.33% / **8.36** -45.03% |
| NN (Base) | 85.70 | 97.72 | 5.38 / 8.22 | 12.55 / 16.47 |
| NN+Proj | 85.89 +0.22% | 97.76 +0.04% | 2.04 -62.07% / 2.27 -72.39% | 5.52 -56.00% / 6.46 -60.77% |
| NN+Dis | 85.99 +0.34% | 97.78 +0.06% | 1.97 -63.36% / 2.22 -72.98% | 5.34 -57.42% / 6.45 -60.81% |
| SenSR | 84.49 -1.41% | 97.55 -0.18% | 2.58 -51.99% / 3.23 -60.67% | 5.56 -55.68% / 7.81 -52.57% |
| SenSeI | 84.59 -1.30% | 97.49 -0.24% | 7.01 +30.33% / 10.83 +31.64% | 18.22 +45.16% / 24.99 +51.72% |
| LCIFR | 74.53 -13.03% | 95.28 -2.50% | 2.63 -51.05% / 3.06 -62.79% | **3.35** -73.28% / **3.78** -77.07% |
| NN+Anti | 86.11 +0.47% | 97.79 +0.07% | 1.59 -70.38% / 1.94 -76.44% | 4.60 -63.36% / 6.31 -61.69% |
| NN+Anti+Dis | 86.07 +0.43% | 97.79 +0.06% | 1.54 -71.31% / **1.80** -78.05% | 4.44 -64.66% / 5.47 -66.78% |
| DRO-Anti | **86.56** +1.00% | **97.88** +0.16% | **1.52** -71.75% / 1.82 -77.82% | 4.10 -67.34% / 5.54 -66.33% |

Figure 1: Box plots for experimental results on *Compas* dataset. Experiments in the left three figures use Logistic Regression as the base model, and the right three figures use Neural Networks. The top two rows plot the results in individual fairness, while the bottom two rows plot the model's utility. Since we set two sensitive attributes for *Compas* dataset, we plot three situations for *comparable samples* upon sensitive attributes for these two samples, and use logical expressions to denote them. We use 'and' to indicate none of the sensitive attributes is same between a pair of *comparable samples*, use 'or' to denote at least one sensitive attribute is different, and use 'not' to indicate both two sensitive attributes are consistent. The dash line in the box plots indicate the arithmetic mean.

iterative optimization in every epoch to search for good perturbations. In contrast, DRO-Anti omits the inner optimizations but only evaluates every antidote data in each round.

**Binding well with Dis.** Removing sensitive features from input data generally improves individual fairness. In *Law School* dataset, discarding sensitive features can bring up to 44.32% - 63.36% mitigation in individual fairness. But once sensitive features are highly correlated with other features, the mitigation is not guaranteed. In *Adult* dataset, removing sensitive features only gets 0.93% - 2.94% improvements across these two models. Regardless of the varying performance from Dis, our antidote data bind well with sensitive features discarding. On *Adult* dataset, our LR+Anti plus

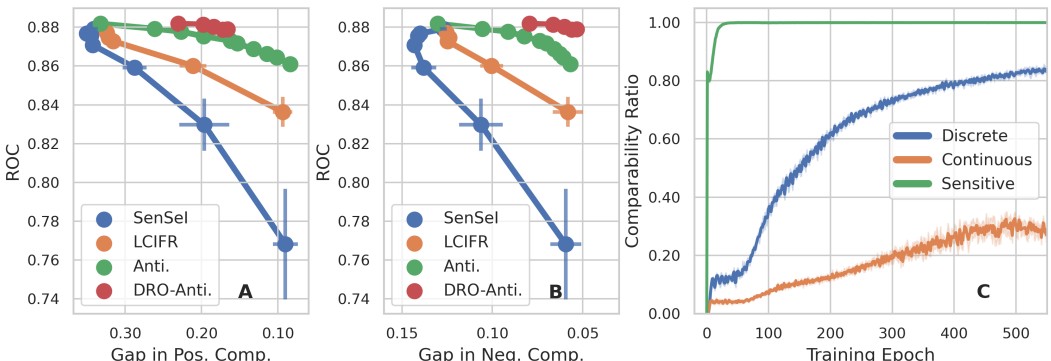

Figure 2: **A & B**: The tradeoffs between utility and fairness on *Adult* dataset. For SenSeI we iterate the controlling hyperparameter in (1e+3, 5e+3, 1e+4, 5e+4, 1e+5, 2e+5, 5e+5). For LCIFR, we iterate the weight for fairness in (0.1, 1.0, 10.0, 50.0, 100.0). For Anti, we have the proportion of antidote ratio at 0%, 45%, 90%, 134%, 180%, 225%, 270%, 316%, 361%, and 406%. For DRO-Anti, we have the proportion of antidote ratio at 45%, 90%, 136%, 180%, 225%. Every point is plotted with variances, and the variance for our models is too small to observe in this figure. **C**: The convergence in terms of the comparability ratio during the training of the generator.

Dis boosts individual fairness in Pos. Comp. by 5.36%, where solely discarding sensitive features only has 0.94% improvements. This number is consistent in NN, *i.e.*, 4.96% compared to 0.93%.

**Algorithmic Tradeoffs** In Figure 2 **A & B**, we show the tradeoffs between utility and fairness. We have two major observations: **(1)** Models with antidote data perform better tradeoffs, *i.e.*, with more antidote data, we have lower individual unfairness, and less drop in model utility. DRO-Anti has the best tradeoffs and achieves individual fairness with an inconspicuous sacrifice of utility even when the amount of antidote data goes up. **(2)** Our models enjoy a lower variance with different random seeds. For baseline methods, when we turn up the hyperparameters controlling the tradeoffs, there is an instability in the final results and a significant variance. However, as our model is optimized on approximately real data, and with no change on a model from Anti. and minimal change in optimization from DRO-Anti, there is no observational variance in the final results.

**Convergence** In Figure 2 **C**, we show the change of the comparability ratio, *i.e.*, the rate of comparable samples from the entire generated samples, during training for different types of features. The comparability ratio of sensitive features quickly converged to 1 since we have direct supervision. The ratio of discrete and numerical features converged around the 500-th iteration due to the implicit supervision from the discriminator. The ratio of continuous features is lower than discrete features due to more complex patterns. Due to the imperfect comparability ratio, we add an additional step `Post()` to filter out incomparable samples.

### 4.3 MODELING THE DATA MANIFOLD

**Compare to Randomly Generated Comparable Samples** In Table 3 we compare randomly generated comparable samples to emphasize the benefit of data manifold modeling. We sample the random comparable samples as such: (1) Uniformly sample discrete features and perturb them into a random value in the current feature. The total number of perturbed features is arbitrary in $[0, T_d]$. (2) Uniformly sample values from $[-T_c, T_c]$, and add the per-

Table 3: Comparing to random generated *comparable samples* on *Adult* dataset

| | ROC ↑ | Pos. Comp. (Mean/Q3) ↓ |
|---|---|---|
| NN | 88.18 | 33.21 / 47.84 |
| +100.0% Rand. | **88.25** | 31.33 / 44.69 |
| +200.0% Rand. | 88.18 | 30.16 / 42.51 |
| +300.0% Rand. | 88.19 | 29.48 / 39.77 |
| +500.0% Rand. | 88.08 | 27.94 / 39.31 |
| +44.5% Anti | 87.95 | **26.05 / 35.76** |

turbations to continuous features. We clip the perturbed features in $[0, 1]$. (3) Randomly perturb an arbitrary number of sensitive features. We add these randomly generated comparable samples to the original training set. From the results in Table 3, we observe that with only 44.5% antidote data, the model outperforms the one with 500% randomly generated comparable samples in terms of individual fairness. By surpassing 10x data efficacy, the results demonstrated that modeling on-manifold comparable samples is greatly helpful to mitigate individual unfairness.

**Learning Efficacy of Antidote Data** In Table 4 we study the model binary classification performance by training only on generated data. We use Accuracy (Acc.), Bal. Acc. (Balance Accuracy),

and F1 Score (F1) for evaluation. We construct a synthetic training set that has the same amount of data as the original training set. We use two baselines. *Random Data*: the randomly generated data fit the basic constraints from tabular. *Pert. in SenSeI*: we collect perturbations from the original data in every training iteration of SenSeI, and uniformly sample from these perturbations.

Within expectation, results in Table 4 show that our antidote data suffer from a performance drop compared to the original data because the generator cannot perfectly fit the data manifold. Even so, antidote data surpass random data and perturbations from SenSeI, indicating that antidote data are closer to the original training data.

Table 4: Learning efficacy on *Adult* dataset

|  | Acc. ↑ | Bal. Acc. ↑ | F1 ↑ |
|---|---|---|---|
| Original Data | 84.64 | 76.16 | 65.55 |
| Random Data | 30.48 | 40.25 | 29.59 |
| Pert. in SenSeI | 53.81 | 67.83 | 50.36 |
| Antidote Data | 78.48 | 74.03 | 59.84 |

## 5 RELATED WORK

**Machine Learning Fairness** AI Fairness proposes ethical regulations to rectify algorithms not discriminating against any party or individual. To quantify the goal, the concept 'group fairness' asks for equalized outcomes from algorithms across sensitive groups in terms of statistics like true positive rate or positive rate (Hardt et al., 2016). Similarly, minimax fairness (Hashimoto et al., 2018) characterizes the algorithmic performance of the worst-performed group among all. Though appealing, both of these two notions guarantee poorly on individuals. To compensate for the deficiency, counterfactual fairness (Kusner et al., 2017) describes the consistency of algorithms on one instance and its counterfacts when sensitive attributes got changed. However, this notion and corresponding evaluations strongly rely on the casual structure (Glymour et al., 2016) which originates from the data generating process. Thus, in practice, an explicit modeling is usually unavailable. Individual fairness (Dwork et al., 2012) describes the pair-wise predictive gaps between similar instances, and it is feasible when the constraints in input and output spaces are properly defined.

**Individual Fairness** Several methods have been proposed for individual fairness. Sharifi-Malvajerdi et al. (2019) study Average Individual Fairness. They regulate the average error rate for individuals on a series of classification tasks with different targets, and bound the rate for the worst-performed individual. Yurochkin et al. (2020); Yurochkin & Sun (2021); Ruoss et al. (2020); Yeom & Fredrikson (2021) develop models via DRO that iteratively optimized at samples which violate fairness at most. To overcome the hardness for choosing distance functions, Mukherjee et al. (2020) inherit the knowledge of similar/dissimilar pairs of inputs, and propose to learn good similarity metrics from data. Ilvento (2020) learns metrics for individual fairness from human judgements, and construct an approximation from a limited queries to the arbiter. Petersen et al. (2021) propose a graph smoothing approach to mitigate individual bias based on a similarity graph. Lahoti et al. (2019) propose a probabilistic mapping from input to low-rank representations that reconcile individual fairness well. To introduce individual fairness to more applications, Vargo et al. (2021) study individual fairness in gradient boosting, and the model is able to work with non-smooth models such as decision trees. Dwork et al. (2020) study individual fairness in a multi-stage pipeline. Maity et al. (2021); John et al. (2020) study model auditing with individual fairness.

**Crafting Adversarial Samples** Beyond regular adversary (Madry et al., 2018), using generative models to craft on-manifold adversarial samples is an attractive technique for model robustness (Xiao et al., 2018; Zhao et al., 2018; Kos et al., 2018; Song et al., 2018). Compared to general adversarial samples without too many data-dependent considerations, generative samples are good approximations to the data distribution and can offer attacks with rich semantics. Experimentally, crafting adversarial samples is in accordance with intuition and has shown to boost model generalization capacity (Stutz et al., 2019; Raghunathan et al., 2019).

## 6 CONCLUSION

In this paper we studied individual fairness on tabular datasets, and focused on an individual fairness definition with rich semantics. We proposed an antidote data generator to learn on-manifold comparable samples, and used the generator to produce antidote data for the individual fairness purpose. We provided two approaches to equip antidote data to regular classification pipeline or a distributionally robust optimization paradigm. By incorporating generated antidote data, we showed good individual fairness as well as good tradeoffs between predictive utility and individual fairness.

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

## A  DATASET

***Adult* dataset**  The *Adult* dataset contains census personal records with attributes like age, education, race, etc. The task is to determine whether a person makes over \$50K a year. We use 45.25% antidote data for Anti, and 225.97% antidote data for DRO-Anti. We set $T_d = 1$ and $T_c = 0.025$ for the constraints of *comparable samples*.

***Compas* dataset**  The *Compas* dataset is a criminological dataset recording prisoners' information like criminal history, jail and prison time, demographic, sex, etc. The task is to predict a recidivism risk score for defendants. We use 148.55% antidote data for Anti, and 184.89% antidote data for DRO-Anti. We set $T_d = 1$ and $T_c = 0.025$. Note that from (Bao et al., 2021), *Compas* dataset may not be the ideal dataset for demonstrating algorithmic fairness.

***Law School* dataset**  The *Law School* dataset dataset contains law school admission records. The goal is to predict whether a candidate would pass the bar exam, with available features like sex, race, and student's decile, etc. We use 56.18% antidote data for Anti, and 338.50% antidote data for DRO-Anti. We set $T_d = 1$ and $T_c = 0.1$.

***Oulad***  The Open University Learning Analytics (*Oulad*) dataset contains information of students and their activities in the virtual learning environment for seven courses. It offers students' gender, region, age, and academic information to predict students' final results in a module-presentation. We use 523.23% antidote data for Anti, and 747.85% antidote data for DRO-Anti. We set $T_d = 1$ and $T_c = 0.025$.

***Dutch* dataset**  The *Dutch* dataset dataset shows people profiles in Netherlands in 2001. It provides information like sex, age, household, citizenship, etc., and aim to predict a person's occupation. We remove 8,549 duplication in the test set and reduce the size to 6,556. We use 205.44% antidote data for Anti, and 770.65% antidote data for DRO-Anti. We set $T_d = 1$ and $T_c = 0.025$.

Table 5: Dataset Statistics. We report data statistic including sample size as well as the number of positive and negative comparable samples in training / testing set, respectively.

| Dataset | #Sample | #Dim. | Sensitive Attribute | #Pos. Comp. | #Neg. Comp. |
|---|---|---|---|---|---|
| *Adult* | 30,162 / 15,060 | 103 | marital-status | 739 / 193 | 38,826 / 10,412 |
| *Compas* | 4,626 / 1,541 | 354 | race + sex | 24,292 / 2,571 | 8,116 / 1,020 |
| *Law School* | 15,598 / 5,200 | 23 | race | 13,425 / 1,530 | 1,068 / 118 |
| *Oulad* | 16,177 / 5,385 | 48 | age_band | 33,747 / 3,927 | 5,869 / 608 |
| *Dutch* | 45,315 / 6,556 | 61 | sex | 1,460,028 / 6,727 | 1,301,376 / 9,390 |

## B  IMPLEMENTATION DETAILS

We elaborate the architecture of our model in details by using $h$ as the hidden representations.

$$g_\theta = \begin{cases} h_1 = \text{ReLU}(\text{BatchNorm1d}(\text{Linear}_{\to 256}(\tilde{x} \oplus \tilde{s} \oplus \mathbf{z}))) \oplus \tilde{x} \oplus \tilde{s} \oplus \mathbf{z} \\ h_2 = \text{ReLU}(\text{BatchNorm1d}(\text{Linear}_{\to 256}(h_1))) \oplus h_1 \\ h_3 = \text{ReLU}(\text{BatchNorm1d}(\text{Linear}_{\to \text{Dim}(\tilde{x})}(h_2))) \\ \hat{v}_i = \tanh(\text{Linear}_{\to 1}(h_3[\text{index for } v_i])) \quad \forall\, 0 \le i \le N_c \\ \hat{\mathbf{e}}_i = \text{gumbel}_{0.2}(\text{Linear}_{\to |d_i|}(h_3[\text{index for } K_i])) \quad \forall\, 0 \le i \le N_c \\ \hat{\mathbf{d}}_i = \text{gumbel}_{0.2}(\text{Linear}_{\to |d_i|}(h_3[\text{index for } d_i])) \quad \forall\, 0 \le i \le N_d \end{cases}$$

$$d_\theta = \begin{cases} h_1 = \text{Dropout}_{0.5}(\text{LeakyReLU}_{0.2}(\text{Linear}_{\to 256}(\hat{x} \oplus \tilde{x} \oplus \hat{x} - \tilde{x}))) \\ h_2 = \text{Dropout}_{0.5}(\text{LeakyReLU}_{0.2}(\text{Linear}_{\to 256}(h_1))) \\ \text{score} = \text{Linear}_{\to 1}(h_2) \end{cases}$$

We use Adam optimizer. We set the learning rate for generator $g_\theta$ to 2e-4, for discriminator $d_\theta$ to 2e-4, weight decay for $g_\theta$ to 1e-6, for $d_\theta$ to 0. We set batch size to 4096 and training epochs to 500.

## C  ADDITIONAL RESULTS

We present experimental results on *Dutch* dataset in Table 6, and on *Oulad* dataset in Table 7, and tradeoffs study in Figure 3. Similar conclusions can be drawn as in Section 4.2: with antidote data, our models Anti and DRO-Anti achieve good individual fairness and favorable tradeoffs between fairness and model predictive utility.

Table 6: Experimental results on *Dutch* dataset

|  | ROC ↑ | AP ↑ | Pos. Comp. (Mean/Q3) ↓ | Neg. Comp. (Mean/Q3) ↓ |
|---|---|---|---|---|
| LR (Base) | **89.55** | **87.87** | 17.84 / 24.15 | 21.81 / 30.29 |
| LR+Proj | 86.62 -3.28% | 85.13 -3.13% | 7.74 -56.60% / 8.21 -65.99% | 8.01 -63.30% / 8.55 -71.79% |
| LR+Dis | 87.51 -2.28% | 85.71 -2.47% | 8.44 -52.67% / 9.03 -62.63% | 9.11 -58.22% / 11.29 -62.72% |
| LR+Anti | 85.41 -4.63% | 83.38 -5.11% | 9.55 -46.47% / 10.37 -57.05% | 10.74 -50.77% / 12.70 -58.09% |
| LR+Anti+Dis | 87.40 -2.40% | 85.51 -2.69% | **7.08** -60.32% / **7.06** -70.76% | **7.10** -67.44% / **7.48** -75.31% |
| NN (Base) | **90.22** | **88.93** | 15.88 / 20.85 | 21.42 / 31.68 |
| NN+Proj | 88.18 -2.26% | 86.94 -2.23% | 8.11 -48.95% / 9.44 -54.75% | 7.65 -64.29% / 9.73 -69.28% |
| NN+Dis | 88.21 -2.23% | 86.92 -2.25% | 8.18 -48.51% / 9.41 -54.87% | 8.18 -61.80% / 10.53 -66.76% |
| SenSR | 87.78 -2.70% | 86.68 -2.52% | 8.54 -46.20% / 9.71 -53.46% | 7.72 -63.94% / 8.61 -72.83% |
| SenSeI | 89.91 -0.34% | 88.34 -0.65% | 16.21 +2.07% / 21.12 +1.29% | 21.98 +2.65% / 31.25 -1.35% |
| LCIFR | 88.04 -2.42% | 86.54 -2.68% | 8.12 -48.84% / 9.30 -55.42% | 8.41 -60.73% / 10.61 -66.50% |
| NN+Anti | 87.05 -3.51% | 85.59 -3.75% | 8.71 -45.13% / 10.50 -49.67% | 9.49 -55.70% / 13.23 -58.23% |
| NN+Anti+Dis | 87.80 -2.68% | 86.37 -2.87% | 6.78 -57.30% / 7.37 -64.65% | 6.32 -70.48% / 7.15 -77.44% |
| DRO | 88.00 -2.46% | 87.13 -2.02% | **6.34** -60.04% / **6.06** -70.93% | **4.91** -77.08% / **5.41** -82.93% |

Table 7: Experimental results on *Oulad* dataset

|  | ROC ↑ | AP ↑ | Pos. Comp. (Mean/Q3) ↓ | Neg. Comp. (Mean/Q3) ↓ |
|---|---|---|---|---|
| LR (Base) | 63.04 | 76.73 | 8.41 / 12.09 | 9.08 / 12.97 |
| LR+Proj | **65.20** +3.44% | **79.29** +3.34% | 5.33 -36.61% / 7.50 -37.99% | **5.46** -39.88% / **7.69** -40.71% |
| LR+Dis | 62.52 -0.83% | 76.39 -0.45% | 5.42 -35.50% / 7.89 -34.77% | 5.89 -35.14% / 8.74 -32.63% |
| LR+Anti | 62.17 -1.38% | 76.24 -0.64% | 6.42 -23.61% / 9.19 -24.02% | 7.10 -21.76% / 9.78 -24.63% |
| LR+Anti+Dis | 60.82 -3.52% | 75.07 -2.17% | **5.10** -39.31% / **6.95** -42.53% | 5.81 -36.04% / 8.67 -33.17% |
| NN (Base) | **65.80** | **79.72** | 6.63 / 9.59 | 6.81 / 9.73 |
| NN+Proj | 65.42 -0.57% | 79.49 -0.29% | 4.76 -28.26% / 6.70 -30.11% | 4.65 -31.68% / 6.71 -31.00% |
| NN+Dis | 65.51 -0.43% | 79.59 -0.16% | 4.78 -27.94% / 6.84 -28.75% | 4.75 -30.27% / 6.94 -28.66% |
| SenSR | 65.58 -0.34% | 79.57 -0.19% | 4.96 -25.17% / 7.00 -27.08% | 4.23 -37.85% / 6.16 -36.67% |
| SenSeI | 64.14 -2.52% | 78.68 -1.31% | 5.53 -16.49% / 8.13 -15.22% | 5.50 -19.18% / 7.99 -17.90% |
| LCIFR | 65.21 -0.89% | 79.40 -0.40% | 4.13 -37.61% / 5.66 -41.01% | **3.70** -45.70% / **5.26** -45.94% |
| NN+Anti | 64.75 -1.59% | 79.08 -0.80% | 4.09 -38.32% / 5.82 -39.36% | 4.51 -33.69% / 6.30 -35.27% |
| NN+Anti+Dis | 64.97 -1.26% | 79.20 -0.65% | 4.00 -39.70% / 5.53 -42.33% | 4.18 -38.68% / 5.97 -38.61% |
| DRO | 64.38 -2.16% | 78.62 -1.38% | **2.86** -56.80% / **3.71** -61.34% | 3.97 -41.64% / **5.22** -46.31% |

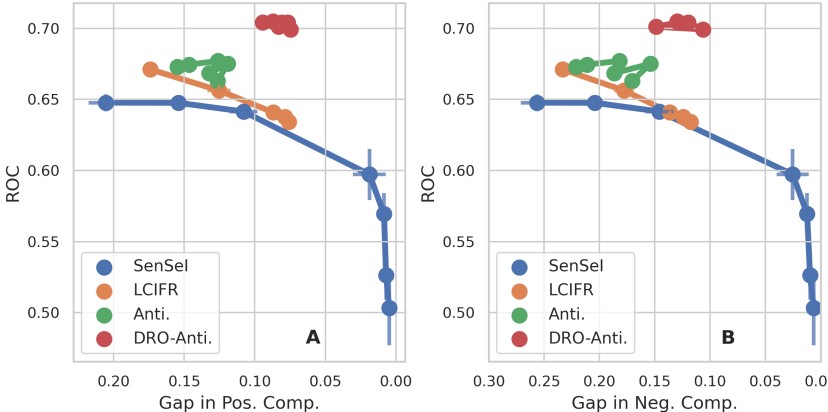

Figure 3: The tradeoffs between utility and fairness on *Compas* dataset. For SenSeI we iterate the controlling hyperparameter in (1e+3, 5e+3, 1e+4, 5e+4, 1e+5, 2e+5, 5e+5). For LCIFR, we iterate the weight for fairness in (0.1, 1.0, 10.0, 50.0, 100.0). For Anti, we have the proportion of antidote ratio at 110%, 130%, 150%, 167%, 185%, 206%. For DRO-Anti, we have the proportion of antidote ratio at 129%, 146%, 167%, 184%, 201%, 222%. Every point is plotted with variances.

