# OpenReview forum: "Learning Antidote Data to Individual Unfairness"
_ICLR.cc/2023/Conference — Submitted to ICLR 2023_

### Official Review · Reviewer_rYKn · 2022-10-21

**Confidence:** 4
**Correctness:** 3
**Technical Novelty And Significance:** 2
**Empirical Novelty And Significance:** 3
**Recommendation:** 3

**Clarity, Quality, Novelty And Reproducibility:**

Clarity: pieces of exposition around comparable data and the generation process aren't clear
Quality: it's a primarily empirical paper and the experiments seem of good quality, and thorough
Novelty: there is some novelty here with respect to the IF+DRO literature I believe, but a fair amount of similar work in a related vein which should at least be conceptually compared against, see FairGAN (https://ieeexplore.ieee.org/abstract/document/8622525) or http://krvarshney.github.io/pubs/SharmaZRBMV_aies2020.pdf
Reproducibility: pretty reproducible, although some hyperparameters not given

**Strength And Weaknesses:**

Strengths:
- generative data approaches to individual fairness questions are sensible and promising
- experimental evaluation is pretty extensive and thorough

Weaknesses:
- The notion of "comparable samples" feels a little bit off to me, and may not capture some important cases. For instance, there could be some non-sensitive feature which nonetheless correlates strong with a sensitive feature, and when trying to find a comparable sample for a given sample, we may have to change that feature by a large absolute amount. However, this type of change is disallowed by Def 2.1, and in my opinion, really restricts the cases that we could apply this model to.
- The setup around the notation and "comparable samples" definition feels a little bit ad-hoc, and doesn't totally define some cases. what if we have ordinal features? are they considered continuous or discrete in this framework? are sensitive attributes s continuous or discrete? In Def 2.1: not sure why the discrete features are constrained by a sum, but the continuous features are constrained by a max
- I don't quite understand all the choices made in the data generating setup. For instance, if we're using GANs I don't understand why we first need to model the continuous data as a mixture of Gaussians - there are some implicit assumptions here that I think should be stated. I'm also not sure why you can't guarantee that your outputs will be comparable according to Def 2.1 by, for instance, clipping the output to the required range.
- some experimental details are a little bit unclear to me - I can't quite follow exactly where only real samples are used, where only generated samples are used, and where a mixture is used. It would be good to have clear delineation of these regimes. Also, it seems that some detail/hyperparameters are missing around the antidote generation - for instance how many comparable samples are generated at each DRO step.
- some confusion in the additional experiments: 1. I can't find the comparability ratio defined anywhere 2. I'm not sure why you would need an additional Post(x) step


Notes:
- the paper discusses IF only in terms of sensitive attributes (see the abstract), but there is a large literature which discusses it without reference to sensitive attributes (e.g. https://arxiv.org/pdf/1905.10607.pdf) - would be good to note this early on, in the introduction
- bottom of p1: "primary use case of DRO in model robustness is to adversarially perturb a sample by a small degree" - don't agree with this statement, there is a long line of work in statistics, recently popular in ML (particularly see John Duchi + collaborators' work eg. https://arxiv.org/abs/1810.08750) which uses a different approach to DRO based on perturbing distributions rather than samples
- the inequality in the last line of Eq 8 loses me, I think a little bit more explanation would help
- in general, I recommend against using the COMPAS dataset - it's not a great dataset for demonstrating points about algorithmic fairness - see https://openreview.net/forum?id=qeM58whnpXM for some commentary
- it would be good to list the hyperparameters which are "default for other methods" in your paper itself, just so it's self contained.
- a graphical representation of Table 1 might be helpful
- Figure 1 and 2 would benefit from having a black-and-white friendly visualization
- a group fairness type baseline - e.g. post-processing, would be interesting here, just to note that it isnt' helpful for your objective
- bottom of p8 - total number of pertubed features should be in (0, T_d)
- bottom of p8 - adding noise from [-Tc, Tc] seems a lot, wouldn't this result in a lot of extreme values?
- the "Learning Efficacy" experiment is interesting, but I'd be very interested to see what happens as you add antidote data to the original dataset

**Summary Of The Paper:**

The authors proposed a data-augmentation method to improve a measure of individual fairness in classification. They propose to learn an "antidote data generator" using GANs, which is intended to learn "comparable" samples to any given sample from another sensitive class. This data could then be added directly to the training data or incorporated using a form of DRO. Experimentally, they show that this provides better results in some individual fairness metrics, without losing too much predictive utility.

**Summary Of The Review:**

I think this paper is probably not quite there in terms of the clarity of the ideas and novelty. Due to these issues, the experimental section, which is pretty thorough, doesn't quite have the impact that I think it could. As such, I'm recommending rejection.

---

> ### Author Response · Authors · 2022-11-16
> **Response to Reviewer rYKn [1/2]**
>
> We are more than cheerful to receive all these insightful comments with detailed explanations and suggestions. Your expertise really helps us improve our paper!
>
> ---
>
> ### 1. The Definition of ‘Comparable Samples’
>
> Our purpose of raising ‘comparable samples’ is to give a **concrete** and **semantically rich** definition of ‘similar samples’ for individual fairness. The definition is highly customizable upon task demand, and until now, to our best knowledge, there is no perfect definition for ‘similar samples’ in individual fairness literature (e.g., previous works define comparable samples using euclidean distance in representational space, which lacks semantic meaning).
>
>  - Feature correlations
> You mentioned that some features might be correlated with the sensitive attributes changed by comparable samples. This is a fact that we totally agree with. However, it could be hard to quantitatively define what is ‘highly correlated,’ and how can this correlation be considered in the definition of comparable samples. If you want to brainstorm any other alternative definition of comparable samples with us, we are totally open to that.
>
>  - Ordinal features
> The current definition is established in an experimental environment and does not consider ordinal features since in our datasets there are no ordinal features. It could be difficult for us to consider a case we haven’t encountered so far. As for an initial solution, ordinal features can be treated as continuous features by normalizing the value into a range [0, 1].
>
>  - Sensitive features
> Sensitive features are discrete in our case, as noted in the ‘Notations’ paragraph on page 2.
>
>  - Constrained by sum/max
> We believe this is a minor issue and the definition can be flexibly defined. Making the continuous feature constrained by a sum or by a max does not affect our method design.
>
> ---
>
> ### 2. Continuous Features with Gaussian Mixture
>
> Modeling the continuous feature using a Gaussian mixture is a choice in generative models for tabular data [1]. It can be treated as a pre-processing approach that first decomposes the complicated distribution of a continuous feature into multiple standard Gaussian distributions for the purpose of easier modeling by a generative model. This technique is demonstrated by previous work [1].
>
> [1]  Modeling tabular data using conditional gan, NeurIPS’19
>
> ---
>
> ### 3. Guarantee of Outputs to be Comparable
>
> It is not guaranteed that a generated output is comparable to an input sample, even with clipping the output to some required range. Consider a case we try to restrict the change of discrete features by at most one feature, but the generated sample gives two features that differ from the input sample. Manually rewriting the features (e.g., choosing one discrete feature to reset) could result in a complicated heuristic.
>
> ---
>
> ### 4. Experimental Details
>
> We are happy to clarify all these details for you and other readers.
>
>  - The usage of real samples & generated samples
> In experiments, LR (base) and NN (base) only use real samples. The method with ‘Anti’ in their name use generated antidote data mixed with real samples. We do not use generated data solely in experiments.
>
>  - How many comparable samples are generated at each DRO step
> Actually, we do not generate comparable samples in DRO, instead, we generate all antidote data used for optimization before the DRO procedure. For example, we use 225% antidote data for DRO on the Adult dataset. All these details are listed in **Appendix A**.
>
>  - Comparability ratio
> Comparability ratio is defined as the ratio of generated samples that satisfy Def. 2.1 from the generator in one iteration of generation using all training samples. We will add this explanation to a proper place.
>
>  - Post(x) step
> This step is to manually select samples from the generator that satisfy Def 2.1 since we cannot guarantee that a generated sample is strictly comparable (See point 3).

---

> > ### Author Response · Authors · 2022-11-16
> > **Response to Reviewer rYKn [2/2]**
> >
> > ### 5. Minors
> >
> >  - Papers on DRO
> > Thanks for pointing these out. We have changed the expression in our updated paper. We shall carefully add discussions to handle the cases raised by these papers.
> >
> >  - Inequality in Eq. 8
> > Explanations added. The inequality is from the definition of comparable samples.
> >
> >  - Compas dataset
> > Thanks for bringing this paper. We will make a note in our paper.
> >
> >  - Hyperparameters for other methods
> > Indeed, we list the hyperparameters used for other methods in the captions of Fig 2 and 3.
> >
> >  - Graphical representation
> > Might be a good idea! However, our concern is, this table contains four metrics, and a graphical representation could be complicated to present all these values, and hard to see the exact number for each method.
> >
> >  - Black-and-white friendly
> > Here we need the color here to indicate additional information in this figure, but would consider your advice seriously.
> >
> >  - Total number of perturbed features
> > The value should be in the range [0, $T_d$], it should include zero feature perturbation and $T_d$.
> >
> >  - Adding noise from [-$T_c$, $T_c$]
> > That won’t be a lot. We mainly set $T_c$ to 0.025 in our experiments, which means for a continuous value in [0, 1], we at most add or minus 0.025 to its current value.
> >
> >  - Learning efficacy experiments
> > We add antidote data to the original dataset through all experimental results in tables, and that is exactly what you are expecting.
> >
> > ---
> >
> > ### 6. Some Other Papers
> >
> > Thanks for bringing these two papers [1, 2]. These two papers use generated data to mitigate group unfairness, and we agree they are conceptually related to our paper. We will add discussion and cite these two papers carefully. For experimentally compared against, we do not find their code for direct implementations.
> >
> > [1] FairGAN: Fairness-aware Generative Adversarial Networks, IEEE Big Data’18
> > [2] Data Augmentation for Discrimination Prevention and Bias Disambiguation, AIES’20

---

> > ### Comment · Reviewer_rYKn · 2022-11-16
> > **Response**
> >
> > Thanks for this rebuttal.
> >
> > On comparable samples: I appreciate the comparison to Euclidean distance as an alternative and agree this is about as good. I guess it puzzles me a little bit - if we are determining in advance that our output will be close to the original sample (on non-sensitive attributes), it's not clear to me how much training a generative model gets us.
> >
> > On definitions: I appreciate the clarifications - I still think the writing of this paper could benefit a little bit by describing the definitions in a bit more generality up front and then making it clear which specific elements are really critical to it.
> >
> > I agree somewhat with reviewer rsds that this reminds me more of adversarial methods rather than DRO, and wouldn’t mind seeing a little bit more about the connection there.
> >
> > In light of this rebuttal, I’ll re-consider my score after engaging in discussion with the other reviewers.

---

> > > ### Author Response · Authors · 2022-11-16
> > > **Thanks for your prompt reply**
> > >
> > > Thanks for your prompt reply in 12h :) We enjoy extending the discussion with you cause we feel you are truly helping to improve our paper instead of just criticizing it.
> > >
> > > ---
> > >
> > > ### 1. Comparable Samples
> > > We are not 100% true to the meaning of '*how much training a generative model gets us.*' If we understand correctly, you are asking what kind of data we are receiving from the generator. In our cases, a generated sample is close to its original samples in terms of non-sensitive features, by the constraints in Def. 2.1., i.e., differs in some discrete or continuous features by some ranges, and has another sensitive attribute. More importantly, the generated data fit the real data distribution but is not an entirely fake one, as modeled by the generative model training on real data distribution. This is not just manually changing the sensitive attributes on one real sample, cause in that case the generated sample could not be a real one, as a motivating example shown on page 3.
> > >
> > > ---
> > >
> > > ### 2. The Definition
> > > We agree generality is good. Our consideration is, a high-level, unified, and maybe more vague definition may not be friendly to readers outside the individual fairness field, if you want to consider all kinds of cases like ordinal features, feature correlations, different constraints, etc. Probably we can add notes in the main paper and extend the discussion around the definition in the appendix to say how can we relax, or modify some constraints to fit other customized demands, does it make more sense to you?
> > >
> > > ---
> > >
> > > ### 3. Adversarial Training and DRO
> > > The term DRO is used in previous individual fairness literature. The idea behind this concept is, we are actively searching for some distribution that best exposes the sensitive information, and the model should be robust/invariant to such perturbed distribution, which results in the name DRO. Adversarial training is like we try to find two samples to confuse some discriminator, and also training the discriminator to distinguish these two samples. We feel like the techniques in the paper are more closer to the concept DRO.

---

> > > > ### Comment · Reviewer_rYKn · 2022-11-17
> > > > **Response**
> > > >
> > > > 1. All I mean is that given some example x, we know that g(x) will lie within a fairly small neighbourhood, so it doesn't seem like the generator is doing that much work, although I guess this depends on the values of T.
> > > >
> > > > 2. This is more of a paper-writing style thing but I prefer seeing an abstract Definition 2.1 and then in a later section explain how the specific cases are handled. To me this is more in line with ML papers I've read, both inside and outside the fairness field. However I can't claim to speak for everyone on this topic.
> > > >
> > > > 3. When I say adversarial training, I mean things like this: https://arxiv.org/abs/2112.08304 - usage of a generator to create adversarial examples which are then added to training. However I am not deeply familiar with adversarial training or this specific line of DRO work in IF.

---

> > > > > ### Author Response · Authors · 2022-11-17
> > > > > **Thanks again for your prompt reply**
> > > > >
> > > > > We appreciate you following up with more comments. We are really glad to address them.
> > > > >
> > > > > ---
> > > > >
> > > > > ### 1. The Contributions of the Generator
> > > > >
> > > > > The purpose of the generator is to enforce the generated antidote data to fit the current data distribution, hence we are expected to get a good tradeoff between utility and individual fairness. There is a good amount of options for generated data originating from Def. 2.1, e.g., perturb which discrete or continuous values, and rewrite with what values, even though the neighborhood looks fairly small at the first glance.
> > > > >
> > > > > Experimentally, in **Table 3, Page 8**, we show how the antidote data coming out from the generator behaves compared to randomly perturbed comparable samples (w.o. the generator). The antidote data from the generator can greatly mitigate individual unfairness with much better data efficacy (44.5% v.s. 500%). We deemed the good results should credit to the generator.
> > > > >
> > > > > ---
> > > > >
> > > > > ### 2. The Definition
> > > > >
> > > > > We would like to invite the reviewer to check **Eq. 1, Page 2** and evaluate if this can be served as an abstraction of Def. 2.1. In our case, all we do is instantiate the distance functions $D_\mathcal{X}$ and $D_\mathcal{Y}$ in a much more semantically-rich form to handle the individual fairness on the experimental tabular data. We agree Def. 2.1 is only one of the forms where individual fairness could happen, but we leave full flexibility to our method to handle various individual fairness definitions as long as there are real exemplars in training data.
> > > > >
> > > > > ---
> > > > >
> > > > > ### 3. Adversarial Training
> > > > >
> > > > > Using generative models to construct adversarial samples is indeed conceptually related to our work. We would love to invite the reviewer to check **the paragraph 'Crafting Adversarial Samples' in the related work section, page 9**, where we discuss the benefit of such adversarial techniques.
> > > > >
> > > > > ---
> > > > >
> > > > > Again, we sincerely thank the reviewer for extending our discussion, and that is where we can learn from a different perspective and strengthen our research capacity. Please let us know whether we address your concerns, thanks!!

---

### Official Review · Reviewer_Q48M · 2022-10-23

**Confidence:** 3
**Correctness:** 3
**Technical Novelty And Significance:** 3
**Empirical Novelty And Significance:** 3
**Recommendation:** 5

**Clarity, Quality, Novelty And Reproducibility:**

Overall, the paper is interesting, but quality and clarity can be improved; please find comments in the above section. There are no new theoretical proofs / techniques.

**Strength And Weaknesses:**

Some comments on strengths of the paper:
1. The paper addresses an important topic of fairness within machine learning.
2. The paper contains good examples of the importance of fairness, as well as good comments on why certain practices in the literature may fail to generate adversarial samples that remain on the data manifold.

Some comments on weaknesses of the paper:
1. Overall, the clarity of the paper can be improved. Many technical details in Section 3 were vague or unclearly defined, see points listed below in 3. for some comments.
2. How are the thresholds Tc and Td chosen for the experiments? Are the results consistent for a wide selection of different threshold values? The numbers shown in Appendix A seem to be quite arbitrary --- are the reported numbers a consequence of choosing a "better" set of thresholds?
3. Some definitions and wordings are unclear and may benefit from a revision. For example:
  - definition 2.1 might benefit from a rephrasing emphasizing "comparable samples given thresholds Td and Tc".
  - In Equation (3), the notions of $\bar{s}$ and $s_{\hat{x}}$ are not immediately clear, although one can trace back to it after reading page
    4 to have a better idea. It may be helpful to include clarifications of definitions early on to aid clarity.
  - The $\oplus$ operator used in (4), (5) and (6) is not explained and causes confusion when understanding the algorithms in page 4.
  - It would also be helpful to be consistent in notation, for example, variations such as "DRO-Anti" and "Dro-Anti" are repeatedly seen.
  - The small term $\delta$ is not clearly defined in (8).
4. What do the experiment numbers look like for other sensitive features? Are there any intuitive explanations to why LCIFR is better in terms of Neg. Comp. on the Law School dataset?
5. It would be helpful to add discussions in the main text to clarify how the generated data using the algorithms of Section 3.1 are "on-manifold": from the paragraph right above Section 3.1, a vague logic/goal of "the generated data should fit into existing data manifold and obey the inherent feature correlations or innate data constraints" is stated. Can this be elaborated? More precisely, why would "comparable samples" fit into "existing data manifold" for general datasets?

**Summary Of The Paper:**

The paper proposes a method to learn and generate "antidote data", which are comparable samples, to resist individual unfairness at minimal cost to model predictivity. The antidote data is generated using a GAN type generator. The paper discusses applications to various datasets and observes the effect of antidote data on fairness for specific sensitive data.

**Summary Of The Review:**

Overall, the paper is interesting but still has shortcomings in its quality and clarity; please find detailed comments in the above sections.

---

> ### Author Response · Authors · 2022-11-16
> **Response to Reviewer Q48M**
>
> Thanks for taking the time to help us improve our paper! We would like to respond to your comments point-to-point.
>
> ---
>
> ### 1. Thresholds $T_c$ and $T_d$
>
> We firmly believe choosing ‘better’ thresholds for the reported numbers should be prohibited from experiments. The thresholds are chosen to guarantee sufficient comparable samples in experiments. We initially set $T_d$ = 1 and $T_c$ = 0.025 and observe these two values can identify enough comparable samples in datasets. These two values are **consistent** through all datasets but are **not dataset-specific** for better experimental results. The only case we enlarge $T_c$ to 0.1 is on Law School Dataset because if $T_c$ is still set to 0.025, we cannot find enough comparable samples for learning and evaluation on that dataset with such strict constraints.
>
> ---
>
> ### 2. Clarity
>
>  - Def 2.1
> The rephrasing you proposed makes sense to us. We have followed your advice.
>
>  - Eq. 3 notations and $\oplus$ operations
> Indeed, we have explanations on those notations in paragraph 2 ‘Notations’, but we guess it may be a litter far from the context which makes you confused here. We have added more explanations here.
>
>  - DRO-Anti and Dro-Anti
> They are typos in our paper. We have made them consistent as ‘DRO-Anti.’
>
>  - $\delta$ in Eq. 8
> $\delta$ in Eq. 8 is the higher-order in Taylor expansion. We have added an explanation in the right place.
>
> ---
>
> ### 3. Other Sensitive Attributes
>
> We switch the sensitive attributes used in the Adult dataset to gender in the following results.
>
> | Method | ROC | AP | Pos. Comp. (Mean/Q3) | Neg. Comp. (Mean/Q3) |
> | :--------- | :----: | :----: | :----: | :----: |
> | LR (Base) | 90.04 | 75.52 | 9.89 / 10.96 | 2.39 / 3.06 |
> | LR + Anti | 90.08 | 75.74 | 9.32 / 10.54 | 1.94 / 2.37 |
> | NN (Base) | 88.18 | 69.99 | 5.86 / 5.92 | 3.06 / 3.60 |
> | NN + Anti | 88.18 | 69.99 | 5.80 / 5.98 | 2.67 / 3.02 |
>
> Due to the complicated patterns in different datasets, unfortunately, we cannot offer a solid hypothesis why one of the competitive methods is better on one dataset in terms of one metric. Instead, we would like to emphasize the overall good performance across multiple datasets.
>
> ---
>
> ### 4. Antidote Data are On-manifold
>
> The argument behind ‘generated data should fit into existing data manifold’ is, we are using generated data for training a classifier, and the classifier is tested on data from the same distribution based on the general ML assumptions. If the generated shifted from the underlying data manifold, the classifier will misfit to a corrupt data distribution, and result in suboptimal performance in evaluation. For example, in models for face verification, we expect all training data to be human faces but not corrupted images or other staff. The good quality of training data will usually bring good performance, and is a consensus in ML.
>
> Experimentally, the good tradeoff between accuracy and fairness from the on-manifold generated data is verified through experiments in Fig. 2 and Fig. 3.

---

> > ### Comment · Reviewer_Q48M · 2022-11-24
> > **Response**
> >
> > Thanks for the responses to my comments in the earlier review. Regarding choices of the thresholds Tc and Td, it might be helpful to show that the results are consistent through a wide range of thresholds. I agree with some points raised by the other reviewers on claims made in the paper, clarity of experiments, and novelty, and have read the corresponding discussions -- I am revising my scores for technical and empirical novelty based on the discussions up to now.

---

### Official Review · Reviewer_rsds · 2022-10-25

**Confidence:** 5
**Correctness:** 2
**Technical Novelty And Significance:** 1
**Empirical Novelty And Significance:** 3
**Recommendation:** 3

**Clarity, Quality, Novelty And Reproducibility:**

I have concerns about the clarity and reproducibility of the work. See comments above.

**Strength And Weaknesses:**


## Major Comments

* The paper is missing discussion of, and comparison against, many of the more canonical works in the DRO literature. It seems to me that all of these should be discussed in the paper (they are not) and more importantly, compared against. These papers present formal approaches to solving the DRO problem and would be important baselines for the current work (which claims to be "similar to DRO" but is in fact not DRO, since there is no notion of worst-case distributional shift being solved for here -- I think the connection to DRO here is not entirely clear, it seems more like adversarial training). For example:

- Sagawa, Shiori, et al. "Distributionally Robust Neural Networks." International Conference on Learning Representations. 2019.

- Levy, Daniel, et al. "Large-scale methods for distributionally robust optimization." Advances in Neural Information Processing Systems 33 (2020): 8847-8860. (This paper has a nice git repo with implementations.)

- Namkoong, Hongseok, and John C. Duchi. "Stochastic gradient methods for distributionally robust optimization with f-divergences." Advances in neural information processing systems 29 (2016).

* Since the paper is explicitly focused on tabular datasets, it should compare against *strong* baselines for tabular data. This is widely considered to be methods such as XGBoost and LightGBM (see e.g. below):

- Shwartz-Ziv, Ravid, and Amitai Armon. "Tabular data: Deep learning is not all you need." Information Fusion 81 (2022): 84-90.

- Grinsztajn, Léo, Edouard Oyallon, and Gaël Varoquaux. "Why do tree-based models still outperform deep learning on tabular data?." arXiv preprint arXiv:2207.08815 (2022).

I will note that *any* of these tabular methods, it seems, could be used alongside "antidote"-augmented datasets, and an augmented dataset should ideally be able to achieve improved fairness without degrading the accuracy of the best possible model.

* A major issue with GAN-style models, and DRO methods, is both are extremely sensitive to hyperparameters. Combining them seems an even more fragile exercise -- please clearly describe the process for hyperparameter tuning/selection (not just the final values, given in B) and what checks were performed to ensure that the results were not sensitive to these parameterizations. This will also be important when the canonical DRO methods described above are used.

* Similarly, it is not clear at all how the percentages of antidote data were chosen: Why do you use 45.25% antitode data for Anti, 225.97% for DRO-Anti on Adult, and 56.18% Angi/338.50% on Law School? How sensitive are the results to these values?


## Minor Comments

* It seems that different sets of results are presented for each dataset (besides the tables). For example, Fig. 1 cocntains extra results for COMPAS, Fig. 2 shows a different set of results for Adult. It would be really nice to have identical plots, for all three datasets, side by side to compare.

* From Figure 1, it looks like none of the methods have significant differences.

* I found the introduction very difficult to read, I would suggest revising for clarity.

## Typos etc.

There are several typos in the paper. Here are a few.

P2: "impracticable"

P2: "may correlated with"

P2: "are come to be comparable"

P3: "by given an original training sample" -> given an original training sample

P5: "censual" - I am not sure this is a word

* The shading in the tables is very difficult to see.

**Summary Of The Paper:**

The paper presents a method for learning a GAN for tabular data that performs conditional generation with semantic constraints for the structural properties in the tabular dataset. This model is used to generate perturbed samples that are, it is argued, intuitively performing a similar role to the perturbations in distributionally robust optimization. These models are created for three standard fairness datasets (Adult, COMPAS, Law School) and empirical results are given.

**Summary Of The Review:**

Tabular data in particular needs more attention in the robustness literature, and I believe in the generative modeling literature as well, and it is good to see the authors' effort in this direction. I have several concerns about the paper in its current form. It seems that only a narrow slice of the DRO literature is discussed, leaving some more mainstream DRO methods out of the work entirely. Well-known effective baselines for tabular data are also not discussed in the work or compared against in the experiments. It is also not clear whether the proposed approach is really equivalent to "robust optimization" at all. The experimental results are a bit difficult to parse due to differences across datasets, but seem inconclusive. Finally, I have some reproducibility concerns. Detailed comments above.

---

> ### Author Response · Authors · 2022-11-16
> **Response to Reviewer rsds [1/2]**
>
> Thanks for all your comments! We would like to address all of your concerns in the followings.
>
> ---
>
> ### 1. Our Method and its Relation to DRO
>
> We are grateful to have so many interesting papers on DRO from your expertise in this domain! We have respectfully read through all these papers. Learning from these papers, we assume the main conceptual gap between your understanding and our motivation is around DRO and its role in our paper. We would like to have the following clarifications on our contributions and the relations between our method and DRO, and we hope it will help us better understand the opinions of each other.
>
> The emphasis of our paper is to **learn antidote data**, and take them in a proper way to mitigate individual unfairness. There are two ways to use the generated antidote data, one is through a simple mixup, and another way is to use a variant of DRO. DRO is only one technique as part of our entire solution for individual unfairness, but not the unique contribution of this paper.
>
> We are **not** trying to propose any novel DRO or improve other DROs. Instead, we are trying to solve the model’s unfairness, and DRO is one technique we adopt for this problem. We would like to respectfully point out that, compared to the literature you listed does not make that sense to us because 1. We are working on individual fairness but they are not, and 2. They are working to improve DRO in various cases but we do not intend to improve that. As for a research paper, we share **completely different goals**, making the proposed comparisons not that straightforward.
>
> ---
>
> ### 2. Other Tabular Data Methods
>
> We agree with your opinion. Our antidote data should work for other tabular methods as well. We present the results obtained from the XGBoost classifier with our augmented datasets on the Adult dataset for your reference.
>
> | Method | ROC | AP | Pos. Comp. (Mean/Q3) | Neg. Comp. (Mean/Q3) |
> | :--------- | :-----: | :-----: | :-----: | :-----: |
> | XGBoost | 92.63 | 82.78 | 10.29 / 15.47 | 11.62 / 18.39 |
> | XGBoost + Anti | 92.57 | 83.01 | 8.89 / 15.27 | 10.52 / 17.51 |
>
> ---
>
> ### 3. Hyperparameters Tuning
>
> Thanks for bringing up this comment.
>
> We borrow the hyperparameters of GAN from [1] and do not further specifically tune them toward any evaluation. For DRO, if you would like to check Alg. 1 on page 5, there is **no hyperparameter** beyond standard neural network training, except the **number of antidote data** used in this stage. For the number of antidote data, we have experimented in **Fig. 2** and **Fig. 3** to show how the performance could be varied by the number of antidote data used in this process. The results show that they are not sensitive to this parameterization.
>
> [1] Modeling tabular data using conditional gan, NeurIPS’19
>
> ---
>
> ### 4. The Percentages of Antidote Data
>
> We would like to first explain where these percentages come from. For one iteration, we input all the training data to the generator and obtain corresponding antidote data for each training instance. We apply the Post() process to filter out generated data that does not satisfy Def. 2.1. This would remain a part of generated antidote data, say, 45.25% antidote data (ratio calculated towards original training data), for this generating iteration. We may generate through multiple iterations (mainly 1 - 8 iterations) to enlarge the number of antidote data since there could be multiple comparable samples for one real sample. The generating would result in many percentage numbers (e.g., 45% - 406% for the Adult dataset).
>
> The variance in datasets brings us different trained antidote data generators for each dataset, and consequently would result in different percentages of antidote data for every generating iteration after going through the Post() process. That is the reason why we have so many non-integer percentages.
>
> The sensitivity of the ratio of antidote data is presented in Fig. 2 A & B and Fig. 3. We would say our method is robust to the number of antidote data in terms of the fairness-accuracy tradeoffs compared to other methods.

---

> > ### Author Response · Authors · 2022-11-16
> > **Response to Reviewer rsds [2/2]**
> >
> > ### 5. Minors
> >
> >  - Different sets of results
> > The origin of different sets of results for these datasets is due to the **number of sensitive attributes** we use in experiments. In the Compas dataset, we use two sensitive attributes so we can visualize the three terms ‘and’ ‘or’ and ‘not’ through figures, while in Adult and other datasets we only use one sensitive attribute so these terms in Compas are unavailable. We choose the sensitive attributes that best expose individual unfairness through classifiers. The comparisons are more meaningful through competitive methods and our approach for each dataset.
> >
> >  - Significant differences in Fig. 1
> > In Fig. 1, the significant difference comes from the upper two subfigures as shown in **‘Gap in Pos. Comp.’** and **‘Gap in Neg. Comp.’**
> >
> >  - Writing of introduction
> > Thanks for bringing that to our attention. Since we do not find any specifically confusing point in the review, we guess probably the sentence ‘Here, ‘similar’ means two instances have close profiles regardless of their different sensitive attributes, and usually have customized definitions upon domain knowledge.’ may bring some confusion to you and readers. We would follow up with an example to best explain the concept of individual fairness here.
> >
> >  - Typos
> > Thanks for pointing out these typos. All fixed.
> >
> >  - Reproducibility
> > We provided all the codes we use for full reproducibility. You are welcome to give it a try, and if you encounter any issues, we are more than happy to help!

---

### Official Review · Reviewer_pdNt · 2022-11-01

**Confidence:** 4
**Correctness:** 3
**Technical Novelty And Significance:** 3
**Empirical Novelty And Significance:** 3
**Recommendation:** 5

**Clarity, Quality, Novelty And Reproducibility:**

Clarity: overall, the writing of the paper is clear.

Quality and novelty: overall, I think the novelty of the paper is moderate.

Reproducibility: Source code is provided.

**Strength And Weaknesses:**

Strengths:
- The technique makes sure that the adversarial samples do not fall far from the data manifold.
- Interesting GAN-based data generators.
- The antidote technique is applicable to both DRO and non-DRO models.
- The experiments demonstrate the benefit of the model compared to baselines.

Weakness:
- The proposed method looks like a variation of data augmentation methods, whereby the samples can be perturbed freely within a boundary without changing the label. The perturbation scheme makes sure that non-sensitive attributes are kept in check, i.e., they should not fall too far from the original data. However, the sensitive attributes can change without any restriction.
To me, this construction looks more similar to a group fairness constraint rather than individual fairness. The groups are represented by the discrete values in the sensitive attributes. The data augmentation scheme suggests the model outputs the same prediction regardless what the value of the sensitive attributes. This is very similar to the concept of demographic parity in group fairness, where the probability of positive prediction should be equalized across multiple groups.
- The metrics chosen for the comparison are different than the metrics used in previous works (e.g., Yurochkin et al. (2020)). For example, the authors use AP rather than the standard accuracy metric for the performance metrics and the positive/negative comparable samples for the fairness metrics. As the 'positive/negative comparable samples' are specifically designed for the 'comparable' concept presented in the paper, this may give an unfair advantage to the proposed model compared to the baselines. I suggest the authors use standard metrics for the evaluation, such as the one used by Yurochkin et al. (2020), for a better comparison with baselines.
- In Table 3, DRO-Anti achieves the worst ROC result. Why is it marked as bold?

**Summary Of The Paper:**

The authors propose a technique for improving distributionally robust techniques for individual fairness by generating antidote samples. These adversarial samples are constrained to be on the manifold of the data distribution. The antidote data generator is a generative adversarial network (GAN) model that takes samples from data distribution and generates comparable samples, where their non-sensitive attributes differ within some thresholded values, whereas their sensitive attributes could differ arbitrarily. The authors then incorporate the generated antidote data into a DRO optimization by forcing the optimization objective not only to optimize the loss w.r.t. original data but also a wors-case loss w.r.t. generated antidote data. Finally, the authors demonstrate the benefit of the model in several public fairness datasets.

**Summary Of The Review:**

It's an interesting paper overall. But I have some concerns about novelty and quality. Therefore, I recommend weak rejection.

---

> ### Author Response · Authors · 2022-11-16
> **Response to Reviewer pdNt [1/2]**
>
> First, we would like to thank you for your insightful reviews. We address your concerns in the following.
>
> ---
>
> ### 1. The Difference Between Group Fairness and Individual Fairness
>
> From the above comments, we believe you are truly an expert in AI Fairness :) We suppose you are talking about the essential **difference** between these two notions: group fairness and individual fairness, **instead of a weakness** of our paper. We are glad to extend our discussion around these two notions with you.
>
> Group fairness is to ensure parity in terms of some statistics across sensitive groups, regardless of the difference in users’ features. Individual fairness, as another notion, says that two samples should receive similar treatments if they are similar in terms of the rest of the features. In other words, group fairness only considers the differences at a group level, while individual fairness considers if two samples are similar, and further more if they have similar treatments. Group fairness does not imply individual fairness, since users in an underrepresented group may suffer from polarization, i.e., some users may receive unreasonable treatments even though the group-level statistics look good.
>
> In our paper, we focus on individual fairness. We consider all sample pairs (we call comparable samples in our paper) that 1. differ in sensitive attributes, and 2. are similar in terms of the rest of the features as defined by Def. 2.1. We construct antidote data based on sample's entire features: not only solely by its sensitive features, but also all the rest of its non-sensitive features to identify individually similar samples. We evaluate the predictive gaps for all comparable samples that are supposed to be individually fair, instead of any group-level statistics.
>
> We hope our explanations can help you and other readers better comprehend these two notions, as well as how our method works on individual fairness. We are open to any further suggestions if we can better and more clearly elaborate these two notions to readers outside this research field.
>
> ---
>
> ### 2. Evaluation Metrics
>
> We are totally open to evaluating using these metrics, and that won’t be a problem at all. Here are the results on the Adult dataset with accuracy.
>
> | Method      | ACC      | ROC  | AP | Pos. Comp. (Mean/Q3) | Neg. Comp. (Mean/Q3) |
> | :------------- | :---------: | :------: | :----: | :-----------------------------: | :------------------------------: |
> | LR (Base)  | 84.64    | 90.04 | 75.72 | 31.75 / 43.55 | 10.25 / 18.37 |
> | LR + Proj   | 80.96    | 81.40 | 62.19 | 25.10  / 34.83 | 23.29 / 33.03 |
> | LR + Dis   | 84.67    | 89.95 | 75.59 | 30.81  / 41.10 | 9.40 / 17.78 |
> | LR + Anti   | 84.21    | 89.72 | 75.04 | 24.72  / 30.84 | 8.66 / 14.64 |
> | LR + Anti + Dis | 84.19    | 89.56 | 74.83 | 23.02  / 26.61 | 8.12 / 13.91 |
> | NN (Base) | 83.28    | 88.18 | 70.09 | 33.21  / 47.84 | 13.03 / 23.37 |
> | NN + Proj | 83.16    | 87.42 | 68.51 | 32.38  / 46.45 | 13.59 / 23.69 |
> | NN + Dis | 83.26    | 88.15 | 70.27 | 32.90  / 44.79 | 11.83 / 23.36 |
> | SenSR | 82.31    | 86.01 | 66.19 | 28.68  / 44.21 | 14.88 / 23.07 |
> | SenSeI | 82.44    | 86.42 | 66.08 | 27.92  / 35.92 | 13.22 / 26.01 |
> | LCIFR | 82.97    | 87.35 | 68.52 | 32.51  / 44.84 | 12.97 / 26.49 |
> | NN + Anti | 83.11    | 87.95 | 69.51 | 26.05  / 35.76 | 10.42 / 16.88 |
> | NN + Anti + Dis | 83.01    | 87.79 | 69.40 | 24.40  / 32.12 | 9.56 / 15.54 |
> | DRO-Anti | 83.02    | 87.91 | 71.08 | 17.46  / 20.04 | 5.48 / 6.87 |
>
> Apart from the above results, we want to explain the **core motivations** behind our evaluations.
>
> 1. Utility metrics. For a probabilistic classifier (output a continuous value typically ranging in [0, 1]) with an undetermined threshold (not necessarily 0.5 but any value), we argue that using ROC and AP is better to describe the entire predictions by skipping thresholding them into binary decisions. It preserves full flexibility to determine the thresholds upon task demands. In our experiments, we consider probabilistic classifiers so we believe AP & ROC is more suitable.
>
> 2. Fairness metrics. We consider the predictive gaps from a probabilistic classifier on two samples holding the same ground truths which are supposed to be individually fair. These samples result in the term ‘positive/negative comparable samples.’ We argue that, under the individual fairness context, comparable samples (differ in sensitive attributes but similar in all the rest of the features) with the same label should receive exactly the same treatments, and that motivates our evaluation and the corresponding concept “positive/negative comparable samples.”
>
> In evaluations, we consider well-motivated and reasonable metrics and procedures, but do not simply follow previous research works without some concrete reasons. If you have a chance to kindly explain why the evaluation metrics used in previous work are ‘standard and reasonable’ we are more than happy to learn :-)

---

> > ### Author Response · Authors · 2022-11-16
> > **Response to Reviewer pdNt [2/2]**
> >
> > ### 3. Bold Issue
> >
> > Fixed. Thanks for pointing that out.
> >
> > ---
> >
> > ### 4. Novelty
> >
> > We notice that you leave an opinion saying the novelty of our paper is ‘moderate’ without any further explanation. To our best knowledge, we are the first paper to use extra data to solve individual unfairness and demonstrate a good fairness-accuracy tradeoff. Would you like to briefly explain the reason behind this opinion? We are eager to learn from your insights, thanks!

---

> > > ### Comment · Reviewer_pdNt · 2022-11-20
> > > **Response**
> > >
> > > Dear authors.
> > > Thank you for the rebuttal.
> > >
> > > **1st discussion point**
> > >
> > > Apologize. I should have stated my comments more clearly. My concern is indeed about the paper's weakness, not the difference between these group fairness and individual fairness. Particularly, even though the method is positioned as a technique for enforcing individual fairness, the consequences of the model resemble more the results of group fairness methods rather than the ones of individual fairness.
> > >
> > > In group fairness models, the groups are usually constructed from the distinct values in the sensitive attributes, for example, the values of sex and racial attributes used in Compass datasets.
> > >
> > > The proposed method indeed utilizes the categorical values of sensitive attributes as a component of the algorithm. Specifically, the antidote data are generated from original samples by perturbing their 'regular' features such that generated samples are still 'nearby' the original sample in terms of their 'regular' features but can vary freely in their sensitive attributes. The method then uses the antidote samples to train the model, either by directly using the antidote data as training samples or by formulating a minimax optimization by selecting the worst-case generated antidote samples.
> > >
> > > The construction above implies that for a particular original sample, the predictions function applied to the nearby areas surrounding the sample should be free from the influence of the sensitive attributes. The antidote data generation force that to happen as it contains generated antidote data with all possible values of the sensitive attributes and it do not change the association between their 'regular' attributes (x = {c, d}) and the label (y).
> > >
> > > Using the group fairness terminology, this can also be stated that group membership should not influence the predictive function, or in a more formal way, P(h(x)|S=s) = P(h(x)|S=s'), \forall s \in S. Here h(x) is the predictive function that takes the regular attributes (discrete or continues), S is the set of possible values of sensitive attributes / group membership, whereas s and s' are a value of the sensitive attributes or the group the sample belongs to.
> > >
> > > The fairness criteria of P(h(x)|S=s) = P(h(x)|S=s'), \forall s \in S is exactly the notion that many group fairness criteria were built on (Agarwal, 2018; Hardt, 2016). In contrast, the individual fairness notion (Dwork, 2012) disregards the concept of groups entirely. Every individual is unique and is treated as such.
> > >
> > > Therefore, after studying the author's proposed technique, I conclude that the technique is secretly a group fairness technique. The connection to the group fairness concept is stronger than to the claimed individual fairness.
> > >
> > > **2nd discussion point**
> > >
> > > Thank you for adding new metrics to the results. Thanks for providing the explanation of why AP and ROC are used instead of the standard accuracy metric. The explanation could be added to the paper. I still see that providing the standard metrics is important for comparability, so I suggest also putting in in the paper/appendix. This will also alleviate possible doubt from the reader, which may have a suspicion that the metrics are carefully chosen to make the proposed method looks better.
> > >
> > > For the fairness metric, I still believe that the metrics used for evaluation should be free from the specific concept introduced in the paper for objectivity and comparability. If another researcher in the future wants to compare her/his new model for individual fairness, the evaluation metrics should also be applicable in that setting.
> > >
> > > **4nd discussion point**
> > >
> > > The evaluation of the 'novelty' of a particular technique should be done by other people, not a self-assessment. I believe my recommendation is in line with the guideline of the ICLR review ("3. The contributions are significant and somewhat new. Aspects of the contributions exist in prior work."). Surely, there are many aspects of the proposed method that exist in prior works, e.g., data augmentation, individual fairness, robust optimization, training procedure, the use of GAN, and so on.
> > >
> > > **Ref:**
> > >
> > > Agarwal, A., Beygelzimer, A., Dudík, M., Langford, J., & Wallach, H. (2018, July). A reductions approach to fair classification. In International Conference on Machine Learning (pp. 60-69). PMLR.
> > >
> > > Hardt, M., Price, E., & Srebro, N. (2016). Equality of opportunity in supervised learning. Advances in neural information processing systems, 29.
> > >
> > > Dwork, C., Hardt, M., Pitassi, T., Reingold, O., & Zemel, R. (2012, January). Fairness through awareness. In Proceedings of the 3rd innovations in theoretical computer science conference (pp. 214-226).
> > >
> > >
> > > **PS**: Sarcasm comments are not necessary for this discourse.

---

> > > > ### Author Response · Authors · 2022-11-20
> > > > **Thanks for your response**
> > > >
> > > > We would like to thank the reviewer for the prompt response.
> > > >
> > > > The relations between individual fairness and group fairness in our paper may come from the realization of 'comparable samples,' i.e., two samples are restricted from some constraints regardless of their sensitive attributes. We found the realization is in a good form to expose individual unfairness in regular classifiers, and still, under the broad concept of individual fairness.
> > > >
> > > > We really appreciate all the comments from reviewers, especially you, as well as the rebuttal procedure. It is a great way to strengthen our research capacity and disconfirm our beliefs from various perspectives. We sincerely hope there are no misunderstandings throughout our discourse and we fully respect your efforts in this voluntary review work.

---

### Author Response · Authors · 2023-02-26
**Paper update**

We want to, again, sincerely thank every reviewer for your valuable comments that help us improve our paper comprehensively.

We have uploaded an updated version of our paper at https://arxiv.org/abs/2211.15897. In this updated version, we improved our paper based on the reviews collected from ICLR'23, including but not limited to 1. discussing and specifying the connections to DRO and its variants; 2. justifying the reasons for the individual fairness notion and setup we consider; 3. describing and clarifying a more detailed experimental setup, and other technical details which might cause confusions; 4. complementing extra experiments on tabular data method.

We cannot improve our paper without generous support from AC and all reviewers. If you have further comments on our paper, please let us know, thanks!

 -- Paper authors

---

### Decision · Program_Chairs · 2023-01-20

**Decision:**

Reject

**Justification For Why Not Higher Score:**

Reviewers remained unconvinced that their chief concerns were addressed, despite author rebuttals.

**Justification For Why Not Lower Score:**

N/A

**Metareview: Summary, Strengths And Weaknesses:**

The paper proposes methodology for achieving individual fairness, based on creating so-called 'antidote' data. Unlikely approaches that aim to achieve insensitivity to sensitive features, which the authors claim results in a focus of the performance of the classifier off-manifold, the approach proposed here attempts to use GANs to generate comparable samples as an antidote to unfairness.

The abstract presents an exciting sounding idea. The reviews were, however, unanimously against publication. I suspect that the writing is mostly to blame in the sense that, had the authors anticipated these points of confusions / counter-arguments, they could have sharpened their paper / experimental work.

Chief among these points of confusions / concerns are:

1. Concern that the approach is actually a group fairness approach. These seems like it should be a straightforward one to address. The reviewer is concerned that the approach here actually introduces group fairness. Of course, that does not imply that it does not also achieve individual fairness. The authors could demonstrate that the approach does not achieve group fairness for a specific reason. But really the authors should focus on rigorous / formal notions of individual fairness and they should provide guarantees or at least evidence that, irrespective of whether group fairness is also achieved, individual fairness is achieved.

2. Concern that the baselines for tabular data need to be refined. The authors quickly produced tabular results for more SOTA methods, but there were precious few details on how these numbers were achieved. The gap also shrank and so this opens up the possibility that, after appropriate tuning of these methods, the gap might vanish. Would that mean the approach is doomed? Not necessarily, but the authors have to at least understand what's going on to tell a different story.

3. Concern that the connection with the proposed method and DRO is not clear. One question here is, what is the formal connection with DRO? Can this approach be understood as precisely a form of DRO? This might be easy to answer if the question is "yes". It might be tricky to argue that the answer is "no". So I'm not certain this is a fair question to demand an answer to before publication. If the connection were better understood, then appropriate baselines / comparisons would be easier to identify. Even if the method is an example of DRO, in practice, there are only a handful of DRO methods and this one seems to be new.  It also seems to be similar to adversarial training. At the end of the day, I think having benchmarks comparing it to the "standard" DRO methods and "standard" adversarial training can serve as initial evidence that there's something new here. If the authors cannot make a formal connection or cannot prove that their method is not an example of DRO / adversarial training, then they might anticipate this and raise this as an open question. (Sometimes the best defense is a strong offense / conjecture.)

In summary, this paper seems to be describing something interesting, but reviewers got too caught up in multiple details that "weren't quite right" in their eyes. I'd recommend a careful revision, anticipating what issues will snag reviewers and distract them from seeing the merits of the work.

**Summary Of Ac-Reviewer Meeting:**

N/A